# Risk Assessment of Nine Coccidiostats in Commercial and Home-Raised Eggs

**DOI:** 10.3390/foods12061225

**Published:** 2023-03-13

**Authors:** Rui R. Martins, André M. P. T. Pereira, Liliana J. G. Silva, Alexandra Esteves, Sofia C. Duarte, Andreia Freitas, Angelina Pena

**Affiliations:** 1LAQV, REQUIMTE, Laboratory of Bromatology and Pharmacognosy, Faculty of Pharmacy, University of Coimbra, Polo III, Azinhaga de Santa Comba, 3000-548 Coimbra, Portugal; 2Centre of Studies in Animal and Veterinary Science (CECAV), University of Trás-os Montes e Alto Douro (UTAD), Apartado 1013, 5001-801 Vila Real, Portugal; 3Centro de Investigação Vasco da Gama, Escola Universitária Vasco da Gama (EUVG), Av. José R. Sousa Fernandes 197, Campus Universitário de Lordemão, 3020-210 Coimbra, Portugal; 4National Institute for Agricultural and Veterinary Research (INIAV), I.P., Av. da República, Quinta do Marquês, 2780-157 Oeiras, Portugal; 5Associated Laboratory for Green Chemistry of the Network of Chemistry and Technology, REQUIMTE/LAQV, R. D. Manuel II, Apartado 55142, 4051-401 Porto, Portugal

**Keywords:** coccidiostats, eggs, UHLC-MS/MS, commercial vs. home-raised, synthetic, ionophore

## Abstract

The poultry industry, in order to prevent and control coccidiosis caused by *Eimeria spp.*, widely uses coccidiostats as feed additives. The main objective of this study was to evaluate the presence of nine coccidiostats in 62 egg samples by UHPLC-MS/MS. Overall, detection frequency and average concentration were 90.3% (56/62) and 106.3 μg kg^−1^, respectively. Only diclazuril and nicarbazin were detected. Diclazuril, only found in home-raised eggs, showed an overall detection frequency of 8.1% (5/62), with average and maximum concentrations of 0.46 ± 1.90 μg kg^−1^ and 13.6 μg kg^−1^, respectively. Nicarbazin presented an overall higher frequency, 88.7% (55/62), with levels up to 744.8 μg kg^−1^. Additionally, four samples (6.5%) presented both nicarbazin and diclazuril. Home-raised egg samples (*n* = 28) showed a detection frequency of 89.3%, with nicarbazin found in more samples (85.7% vs. 17.9%) and at higher levels (266.3 ± 169.4 μg kg^−1^ vs. 0.91 ± 2.78 μg kg^−1^) when compared to diclazuril. In supermarket samples (*n* = 34), only nicarbazin was detected in 31 samples (91.1%), with an average of 167.6 ± 62.2 μg kg^−1^. Considering the average contamination scenario, consumers’ health should not be adversely affected by egg consumption. In every scenario considered, children were the most vulnerable population group.

## 1. Introduction

Chicken eggs have always been part of human diets throughout history. Food and Agriculture Organization (FAO) reports indicate a sharp increase in egg consumption since 2000 (ca. 50%) [1]. In response to their widespread consumer acceptance, increased individual consumption, and growing population, hen eggs and their products are expected to continue to be in demand for a long time to come. [2]. China is the world’s biggest producer of fresh eggs, followed by the United States, India, Japan, Russia, and the European Union (EU). Japan is the country with the highest egg consumption *per capita*. The production of eggs in Portugal exceeds 1.5 million each year, and its consumption *per capita* is 9.8 kg/year [3].

Aside from providing animal protein, eggs also contribute to our daily mineral, vitamin, and fatty acid needs [4]. From a functional point of view, it is particularly interesting that eggs offer a moderate calorie source (about 150 kcal/100 g) and great culinary versatility. With a low price, they are affordable to most people [5], and therefore assume a very important component of a healthy diet [6].

Coccidiosis is a parasitic infection of the intestine caused by protozoa of the phylum Apicomplexa, family Eimeriidae [7]. There are seven *Eimeria* species known to cause serious clinical diseases in poultry, causing bloody droppings, weight loss, and mortality, especially in chicks, because of the rapid parasite replication in host cells (4–7 days). 

Poultry farming has integrated industrial production principles since the 20th century. Technological advancements have led to increased efficiency through improvements in mechanical equipment and advances in genetics, allowing for larger aviaries to house more birds [3]. However, in these crowded conditions, fecal-oral infections spread faster, favoring the small areas and warm, humid environment in which animals are housed [8]. Therefore, nowadays, the production of food of animal origin is largely supported by the adoption of veterinary medicines and feed additives [6].

There are several strains that can cause coccidiosis, which means that the immunity to one strain is not enough to prevent the infection. Additionally, since coccidiostats only inhibit, reversibly, some of the life stages, they must be used prophylactically during their lifetime to prevent infection. Consequently, in order to prevent costly outbreaks of coccidiosis, today’s poultry farmers prophylactically administer coccidiostats in their feed [9]. Thus, coccidiostat residues may be retained in muscle [10] and, eventually, in eggs due to carryover [11], since their use is not permitted in laying hen feed due to the risks of consumer exposure to adverse residue levels [12]. Currently, there are eleven coccidiostats approved for use as poultry feed additives in the European Union (EU), six of which are ionophores and five are synthetic [13]. 

In terms of livestock quality and quantity, coccidiostats offer key advantages through disease prevention, which may lead to their overuse in the agri-food industries. This can be harmful to consumers [14], affecting physiological functions like drug resistance, hypersensitivity, poisoning, carcinogenicity, and teratogenicity [15]. Consequently, legislative bodies set maximum residue limits (MRLs) for a range of residues, including coccidiostats in eggs, adapting new limits to current scientific and technological data to ensure the safety of foodstuffs offered to consumers, ranging between 2 μg kg^−1^, for diclazuril and narasin and 300 μg kg^−1^, for nicarbazin [16,17]. 

Therefore, intensified research and vigilance regarding coccidiostat residues in food are increasingly justified, since these compounds may pose significant health risks, namely to susceptible populations, such as children, due to their physiological immaturity and the higher relationship between body mass and food consumption [18]. 

Given the above, nine ionophores and synthetic coccidiostats (lasalocid, narasin, salinomycin, monensin, maduramicin, halofuginone, robenidine, diclazuril, and nicarbazin) were surveyed in Portuguese eggs, comparing differences between providers, namely supermarkets and domestic producers. As part of this study, an exposure and risk assessment of different population groups, namely children, teenagers, and adults, was also accomplished.

## 2. Materials and Methods

### 2.1. Sampling

From May to July 2022, egg samples were collected from supermarkets and from home productions located in the north and center of mainland Portugal. A total of 62 samples were analyzed: a total of 28 from domestic producers (two from each one) and 34 from supermarkets (two from each brand). After collection, the eggs were opened and poured into a cup. The whole egg (without the eggshell) was homogenized and, afterwards, frozen at −18 °C until analysis. Data on sampling time, location, and producers’ information were recorded for each sample.

### 2.2. Chemicals, Reagents, and Standard Solutions 

The coccidiostat standards acquired, with purity ≥98%, were as follows: lasalocid A sodium salt (LSC), narasin (NRS), salinomycin sodium salt (SLM), monensin sodium salt (MNS), maduramicin ammonium (MDM), halofuginone hydrobromide (HFG), robenidine hydrochloride (RBD), diclazuril (DCZ), 4,4′-dinitrocarbanilide (DNC) (1,3-bis(4-nitrophenyl)urea) as residue marker of nicarbazin, DNC d8, and nigericin sodium salt (NIG). All analytical standards were obtained from Sigma Chemicals Co. (St. Louis, MO, USA). Acetonitrile (ACN) was also obtained from Sigma Chemicals Co. (St. Louis, USA). Bi-distilled water was daily obtained through a Milli-Q system (Millipore, Bedford, MA, USA). Formic acid and dimethyl sulfoxide (DMSO) were purchased from Merck (Darmstadt, Germany).

Standard stock solutions were prepared, at 1 mg mL^−1^, by dilution of the accurate amount in the appropriate solvent (depending on the compound solubility): LSC, NRS, SLM, MNS, MDM, and NIG with ACN; RBD, DCZ, DNC, and DNC d8 with DMSO; HFG with a solution of ACN:H_2_O (50:50). The relevant dilutions, in ACN, were performed in order to obtain a final working standard mixture in accordance with the MRLs established for coccidiostats. The matrix calibration curve was prepared by spiking blank material at five levels, including the blank, ½MRL, 1MRL, 1.5MRL, and 2MRL. The same levels of concentration were used to prepare a standard calibration curve in solvent (mobile phase A). A working solution for the internal standards (IS) was also prepared by performing the necessary dilutions to obtain a solution with 1 µg mL^−1^ of NIG and DNC d8. All standard solutions were stored at −25 °C for a maximum of 12 months.

### 2.3. Sample Extraction

Approximately 3 g of previously defrosted hen eggs were vortex extracted with 10 mL of acetonitrile for 5 min. After a 10 min rest, the sample was placed in an ultrasound bath for 5 min. Subsequently, it was stirred for 10 min on a vertical shaker (Agitelec, J. Toulemonde, Paris, France) and centrifuged (3-16K, SIGMA, St. Louis, MO, USA) for 15 min at 5444× *g* at 4 °C. Afterwards, the supernatant was transferred to a test tube, and the extract was evaporated to dryness at 45 °C under a gentle stream of nitrogen.

### 2.4. UHPLC-MS/MS Analysis

The detection and quantification of coccidiostats was performed with an UHPLC coupled with a triple quadrupole mass spectrometer, namely, a Nexera X2 Shimadzu UHPLC connected to a QTRAP 5500+ (Sciex, Foster City, CA, USA). 

Regarding the chromatography separation conditions, the column used was a Phenomenex Kinetex biphenyl (1.7 µm, 100 A, 2.1 × 50 mm), maintained at 40 °C. The injection volume was 10 μL, and the autosampler was kept at room temperature. The flow rate was 500 μL min^−1^, and the mobile phase consisted of 0.1% formic acid in water (A) and ACN (B). The gradient program was as follows: 0–6 min from 100% A to 100% B and kept until 9 min, then back to 100% A from 9 to 10 min, for a total run time of 11 min. 

In terms of mass spectrometry, the detector was equipped with an electrospray ionization source, simultaneously working in negative and positive mode (ESI+ and ESI−) at 500 °C. The acquisition was performed in multiple reaction monitoring (MRM) with the software Analyst TF (SCIEX, Foster, CA, USA) and the data analysis with MultiQuant (SCIEX, Foster, CA, USA). The conditions optimized for each compound and the correspondent IS are presented in Table 1.

### 2.5. Statistical Analysis

A complete statistical analysis was performed using GraphPad Prism (8.4.3, GraphPad Software, Inc., San Diego, CA, USA). To test whether the dataset was of Gaussian distribution, the D’Agostino–Pearson normality test was used. Since most of the data set was not normally distributed with non-homogeneous variances, nonparametric tests were applied. For the evaluation of three or more data sets, the Kruskal–Wallis test and Dunn’s post-test were used to assess statistical differences. For the comparison between two data sets, the Mann–Whitney test was performed. To calculate coccidiostat averages with concentrations below the limit of detection (LOD), half the LOD was used, and for those with concentrations below the limit of quantification (LOQ), half the LOQ was used. The statistical significance level was set to *p* < 0.05.

### 2.6. Risk Assessment

The estimated daily intake (EDI) was calculated using a deterministic method. The calculation of the EDI was based on Equation (1) [19]:EDI (mg kg^−1^ day^−1^) = (Ʃc) (CN^−1^ D^−1^ K^−1^)(1)
where Ʃc is the sum of each coccidiostat in total samples (mg kg^−1^), C is the estimated annual egg consumption per person, N is the total number of samples, D is the number of days in a year, and K is the mean human body weight (kg). Regarding consumption, a value of 9.8 kg/year *per capita* was considered [3]. Three age-based population groups were considered for the exposure assessment: children (3–9 years old), teenagers (10–17 years old), and adults (18–64 years old). The body weight considered for each group was 25, 55, and 70 kg, respectively [20].

The Joint Food and Agriculture Organization of the United Nations/World Health Organization Expert Committee on Food Additives (JECFA) established acceptable daily intakes (ADIs) for all authorized coccidiostats: a limit of 0.03 mg kg^−1^ of body weight (bw) day^−1^, for diclazuril [21] and 0.9 mg kg^−1^ bw day^−1^ for nicarbazin [22].

## 3. Results and Discussion

### 3.1. Method Validation

A complex matrix such as eggs requires careful and efficient sample preparation. The egg white has a water content of 88.5%, while the egg yolk presents 50%. There is also 33% fat in the yolk, which may contribute to ion suppression during analysis and a non-reproducible signal [19].

In order to demonstrate that the analytical method was suitable for its purpose, it was validated in accordance with Commission Implementing Regulation (CIR) 808/2021 [20] regarding the performance of analytical methods and the interpretation of results. In this process, different parameters were evaluated regarding the requirements for confirmatory methods, and the validation results are summarized in Table 2, namely: CCα, trueness (or recovery), precision, selectivity and specificity, ruggedness, and linearity. All calculations were made using relative areas (the ratio of the area of each analyte to the corresponding internal standard). For positive and negative ionization, nigericin and DNC d8 were used as internal standards, respectively.

Bearing in mind that the analyzed coccidiostats have established MRLs, the CCα calculation considered that it had to be higher but as close as possible to the MRL. As stated in the CIR 808/2021, Equation (2) was used, where σMRL is obtained through the reproducibility achieved by analyzing at least 20 blank samples spiked at the MRL level. The CCα of each compound corresponds to the value above which the sample is considered non-compliant and the food product is regarded as not safe for consumers.
(2)$$\mathrm{CC}\alpha =\mathrm{MRL}+1.64\times \sigma_{\mathrm{MRL}}$$

Although the LOD and LOQ are not included in the validation procedures as described in the CIR 808/2021, they were also calculated in order to have a full understanding of the method’s lower detection levels, regardless of the established MRLs. The determination of both LOD and LOQ was based on the ICH guidelines [21], as shown in Equations (3) and (4), respectively.
(3)$$\mathrm{LOD}=\frac{3.3\times \sigma }{S}$$
(4)$$\mathrm{LOQ}=\frac{10\times \sigma }{S}$$
where σ represents the standard deviation associated with the 20 blank samples analyzed, and S is the slope of the calibration curve. Narasin was the compound with the lowest LOD, 0.1 μg kg^−1^, while lasalocid had the highest value, 2.79 μg kg^−1^. LOQ values vary between 0.20 μg kg^−1^ and 3.11 μg kg^−1^, for diclazuril and lasalocid, respectively. The analysis of those 20 blanks was also used to evaluate the selectivity and specificity. The chromatograms of all compounds in the blank samples were found to be free of any interference in the expected retention time of the targeted coccidiostats.

Depending on the values achieved for the LOD and LOQ, the levels of the spiked replicate were selected to evaluate the recovery and precision. In fact, according to the above-mentioned regulation, the suggested levels are 0.1, 1, and 1.5 MRL. However, in cases where the 0.1 MRL is not achievable, the level of 0.5 MRL is acceptable. In practice, to assess recovery and precision, spiked blank samples at 0.1, 0.5, 1, and 1.5 MRL were analyzed, but for compounds with LOD higher than the 0.1 MRL, the replicates were performed at 0.5, 1, and 1.5 MRL (levels are presented in Table 2). Those replicates were performed (*n* = 6) on three different days to evaluate the robustness of the method. 

The range of recoveries achieved was between 84.4% for lasalocid at 1.5 MRL and 116.1% for the same compound at 0.1 MRL. The acceptance criteria, directly related to the concentration, give the range between 80 and 120% as the stricter criteria, which was fulfilled for all compounds and their corresponding validation levels. Furthermore, concerning the precision evaluation and the requirements of the regulation that defines the Horwitz equation to be used to assess the maximum acceptable precision, it depends on the compound concentration. In that sense, maduramicin was the compound with the worst precision results. The obtained repeatability of 20.6% for 1.5 MRL is below the acceptance value of 16.7%. On the other hand, despite the highest value achieved, the reproducibility of 22.2% for 1.5 MRL of maduramicin is below the acceptance value of 25%. These results suggest that maduramicin may be more affected by interferences in the matrix, leading to higher variability. 

Linearity was evaluated using matrix-matched calibration curves with five concentration levels. As can be observed in Table 2, all calibration curves presented correlation coefficients (R^2^) greater than 0.99.

### 3.2. Occurrence

Out of the 62 samples tested for coccidiostats, 56 (90.3%) were found positive, with an average concentration of 106.3 μg kg^−1^. Figure 1 and Table 3 present the frequency, range, and contamination means of the two coccidiostats found: nicarbazin (an ionophore) and diclazuril (a synthetic) in home-raised and supermarket samples. 

Diclazuril presented a detection frequency of 8.1% (*n* = 5), all in home-raised eggs, with a total average of 0.46 ± 1.90 μg kg^−1^ and a maximum concentration of 13.57 μg kg ^−1^.

Nicarbazin, which appeared in both sample types, presented the highest contamination frequency of 88.7% (*n* = 55), with mean and maximum values of 212.1 ± 131.3 μg kg^−1^ and 744.8 μg kg^−1^, respectively. There was statistical significance (*p* < 0.0001) between diclazuril and nicarbazin.

Nicarbazin and diclazuril were simultaneously found in four home-raised samples (6.5%). In light of these findings, it appears that a combined treatment is a common practice among Portuguese breeders, possibly to maximize the efficiency of treatment and avoid the development of anticoccidial drug resistance. On the other hand, 21.0% (*n* = 13) of the samples contained one or both compounds at levels higher than the established MRLs [17,23].

As shown in Figure 1b and Table 3, home-raised samples showed a total detection frequency of 89.3% (25 out of 28), being contaminated with both diclazuril and nicarbazin. As for supermarket samples, the detection frequency was 91.1% (31 out of 34 samples). While home-raised eggs showed average coccidiostat levels of 133.6 ± 178.9 μg kg^−1^, up to 744.8 μg kg^−1^, supermarket samples presented lower averages of 83.8 ± 95.0 μg kg^−1^ and maximum levels of 222.4 μg kg^−1^.

In supermarket samples, nicarbazin was the only coccidiostat recorded, at 167.6 ± 62.2 μg kg^−1^, up to 222.4 μg kg^−1^. In home-raised eggs, both diclazuril and nicarbazin were found, with maximum concentrations of 13.6 μg kg^−1^ and 744.8 μg kg^−1^, respectively.

The increased concentrations in home-raised samples presented statistical significance (*p* = 0.0281) in comparison with supermarket samples.

There are many known risk factors associated with the cross-contamination of non-target feed and the detection of coccidiostat residues in eggs. These include inadequately cleaning feed reservoirs and equipment, illegally adding coccidiostats to feed, and disrespecting the coccidiostat withdrawal periods established by the European legislation, mostly by domestic breeders [24].

Overall, comparing with the few other previous scientific studies (Table 4), our results showed higher frequencies and higher concentration values. In many cases, the comparison of results is difficult due to differences between EU and other countries regulations, control scopes, and laying hen rearing methods within EU and non-EU countries [2].

Between 2007 and 2010, a study was conducted in Poland to control coccidiostat residues. Among the 312 analyzed eggs, 23 were positive and 14 (4.49%) were found to be non-compliant. As in the present study, the ionophore coccidiostats lasolacid, maduramicin, salinomycin, and semduramicin were the most frequently determined, ranging between 6.3 μg kg^−1^, for salinomycin and 320 μg kg^−1^, for lasalocid [25].

In central Italy, between 2012 and 2017, 151 samples of eggs were also tested for coccidiostats. From these, 15.9% (*n* = 24) were positive for maduramicin, salinomycin, robenidine, diclazuril, and decoquinate. Violative residues were found in 2% (*n* = 2). These non-compliant samples showed a concentration oscillating between 1.2 μg kg^−1^ of diclazuril and 1002 μg kg^−1^ of lasolacid [26].

In the monitoring of 18 coccidiostats, performed in Latvia [27], only 1 egg (about 1%) from the 81 samples analyzed presented a non-conforming result of 102 ± 7 μg kg^−1^ of toltrazuril sulfone. This sample also presented 10.2 ± 0.5 μg kg^−1^ of toltrazuril, 4.0 ± 1.1 μg kg^−1^ of toltrazuril sulfoxide, and 10.1 ± 4.1 μg kg^−1^ of nicarbazin [27]. 

González-Rubio et al., in 2020, in Spain, also investigated narasin, salinomycin, lasalocid, maduramicin, monensin, and semduramicin in eggs. Despite the low number of samples (*n* = 14) tested, they found that only one was positive for narasin [28].

From 2018 to 2020, in Serbia, 255 egg samples were also analyzed for coccidiostats. A total of 56 (22%) were non-compliant with Serbian regulation regarding the presence of robenidine, nicarbazin, salinomycin, maduramicin, and lasolacid, which were found in concentrations ranging between 0.002 and 17.50 μg kg^−1^ [2].

In 2021, 90 Bosnian egg samples were analyzed, and 10 were found positive for lasolacid residues, with concentrations ranging from 7 to 33 μg kg^−1^ [24]. As a poultry exporter to the EU market, Bosnia and Herzegovina must control and monitor the presence of coccidiostats in animal feed and poultry products considering the regulations in force in the EU.

In 2020, EFSA reported 17 non-compliant results (0.35%) for anticoccidials in eggs from six countries. The coccidiostats reported were: diclazuril (3) in Croatia and Slovenia; lasalocid (3) in France and Poland; monensin (3) in Poland; narasin (3) in France and Poland; robenidine (1) in Greece; salinomycin (2) in France and Poland; and toltrazuril sulfon (2) in Latvia [29].

Finally, in the study by Barreto et al. (2017) from Brazil, in which 14 coccidiostats were analyzed in 619 egg samples, only seven showed non-conforming results [30].

**Table 4 foods-12-01225-t004:** Coccidiostats reported in recent studies worldwide.

Country	Year	Coccidiostats	Concentration Range (μg kg^−1^)	Positive/NC Samples	Reference
Poland	2007–2010	Lasolacid Maduramicin Nicarbazin Salinomicyn Semduramicin	280–320 17 16–150 6.3–75 31–180	23/14	[25]
Italy (151)	2012–2017	Lasolacid Nicarbazin Robenidine Diclazuril Decoquinate Salinomicyn Maduramicin	1002 (Lasolacid) - - 1.2 (Diclazuril) - - -	24/3	[26]
Brazil	2017	ND	-	619/7	[30]
Latvia	2019	Nicarbazin Toltrazuril	10.1 ± 4.1 10.2 ± 0.5	5/1	[27]
Serbia (255)	2018–2020	Robenidine Nicarbazin Salinomicyn Maduramicin Lasolacid	0.002–17.50 (all)	69/56	[2]
Spain	2020	Narasin	-	14/1	[28]
Bosnia (90)	2021	Lasolacid	7–33	10/0	[24]
EFSA 4914	2020	Diclazuril Lasolacid Monensin Narasin Robenidine Salinomycin Toltrazurisulfon	-	-/18 (0.35%)	[29]

NC—non-compliant.

Several measures can be adopted for effective coccidiosis prevention and control in order to minimize the disease and production losses. Specifically, management and biosecurity to prevent the introduction of *Eimeria* on the flock; the use of live attenuated and non-attenuated anticoccidial vaccines as an active or passive immunity response to *Eimeria*; or using single drug programs, “shuttle” drug programs, and rotation programs. Even though the use of this specific vaccine is still not widespread, the poultry industry keeps on using concurrent vaccines together with coccidiostat administration [10].

The pattern of results observed in the present study can be explained by the rotational use of coccidiostats. This can also explain the different patterns found between the above-mentioned scientific studies, indicating that coccidiostat occurrence in eggs may fluctuate when different locations and periods are considered [10].

### 3.3. Risk Assessment

Table 5 and Table 6 present the EDI calculation results for each population, based on the average and highest concentrations found, along with the ADI recommended values. Diclazuril presents the lowest ADI, 0.03 mg kg^−1^ day^−1^ [21], followed by nicarbazin, which has an ADI of 0.9 mg kg^−1^ day^−1^ [22].

When assessing the human exposure to coccidiostats through consumption of eggs presenting coccidiostats, based on both average and worst-case scenarios, one can observe that diclazuril and nicarbazin EDI levels were lower than the set ADIs. However, the maximum EDI value was 0.8 mg kg^−1^ day^−1^, obtained for children and nicarbazin when the worst scenario was considered, showing that exposure is close to the ADI of 0.9 mg kg^−1^ day^−1^, with an assessed risk as high as 88.9%. For diclazuril, the highest EDI value was also for children: 0.015 mg kg^−1^ day^−1^, with a calculated risk of 50%. 

When considering the average scenario, the EDI values and risks are, as expected, lower, ranging between 1.76 × 10^−4^ mg kg^−1^ day^−1^ (for diclazuril) and 0.23 mg kg^−1^ day^−1^ (for nicarbazin), and 0.6% (for diclazuril) and 25.5% (for nicarbazin), respectively. 

In every scenario considered, children were the most vulnerable population group, followed by teenagers and adults. In order to evaluate the chronic toxicity of low-level exposure to coccidiostats based on long-term exposure, more research is needed to fully evaluate the risk and potential effects, and special care must be taken with vulnerable groups. 

As part of the European legislation, continuous control over the correct use of coccidiostats is essential to ensure human health protection since, in routine diets, these compounds are also found in other foods, namely poultry meat [31].

## 4. Conclusions

An UHPLC-MS/MS analytical methodology, validated according to Commission Implementing Regulation (CIR) 808/2021, for the determination of nine different coccidiostats (maduramicin, monensin, halofuginone, lasalocid, narasin, salinomycin, robenidine, nicarbazin, and diclazuril), was successfully used in the analysis of 62 chicken eggs acquired from supermarkets and home producers.

Coccidiostats (nicarbazin or diclazuril) were detected in 90.3% (*n* = 56) of the analysed samples. This study demonstrated a higher detection frequency for ionophore compounds. Home-raised samples, with a detection frequency of 89.3% (25 out of 28), were contaminated with diclazuril and nicarbazin. As for supermarket samples, the detection frequency was 91.1% (31 out of 34 samples), being contaminated with nicarbazin only. Four home-raised samples were contaminated with both nicarbazin and diclazuril (14.3%).

Regarding risk assessment, the EDI calculated for nicarbazin in children was 88.9%, when considering the worst-case scenario. However, considering the average contamination scenario, consumers health should not be adversely affected by egg consumption. In every scenario considered, children were the most vulnerable population group, followed by teenagers and adults. Given the scarcity of published studies on this topic, it is imperative to pay greater attention to this topic and expand the sampling.

## Figures and Tables

**Figure 1 foods-12-01225-f001:**
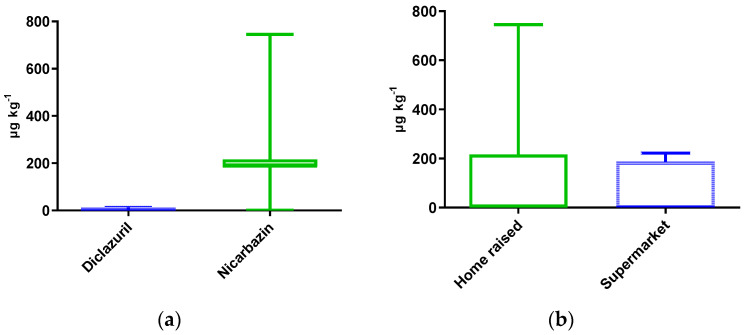
Box and whiskers for the concentrations of the detected coccidiostats diclazuril (blue) and nicarbazin (green) (**a**) and in home-raised (green) and supermarket samples (blue) (**b**).

**Table 1 foods-12-01225-t001:** MRM acquisition conditions for each compound and the corresponding IS.

Compound	ESI	Precursor Ion (*m*/*z* ± 0.5)	Product Ion (*m*/*z* ± 0.5)		Collision Energy (v)	IS
Lasalocid	Positive	613.4	377.3	QP	50	NIG
			359.2	CP	50	
Narasin	Positive	787.6	431.2	QP	69	NIG
			531.5	CP	61	
Salinomycin	Positive	773.5	431.2	QP	69	NIG
			531.4	CP	61	
Monensin	Positive	693.3	675.4	QP	51	NIG
			461.2	CP	67	
Maduramicin	Positive	939.5	877.5	QP	57	NIG
			719.4	CP	89	
Halofuginone	Positive	415.9	138.1	QP	25	NIG
			100.0	CP	55	
Robenidine	Positive	334.0	155.1	QP	29	NIG
			138.0	CP	33	
Diclazuril	Negative	405.0	333.9	QP	−26	DNC d8
		406.9	335.9	CP	−28	
Nicarbazine–DNC	Negative	301.0	137.0	QP	−16	DNC d8
			107.0	CP	−18	
DNC d8	Negative	309.0	141.0	IS	−18	
NIG	Positive	747.1	703.5	IS	−18	

QP—Quantification product ion; CP—Confirmation product ion; IS—Internal Standard; DNC—4,4′-dinitrocarbanilide; DNC d8—Deuterated 4,4′-dinitrocarbanilide; and NIG—Nigericin).

**Table 2 foods-12-01225-t002:** Summary of validation results.

	MRL (μg kg^−1^)	CCα (μg kg^−1^)	LOD (μg kg^−1^)	LOQ (μg kg^−1^)	Validation Level (μg kg^−1^)	Repeatability (RSD%)	Reproducibility (RSD%)	Recovery (%)	Linearity (R^2^)
*Lasalocid*	150	173	2.79	3.11	15 ^(a)^	12.8	14.3	116.1	0.9939
					150	6.1	9.2	91.8	
					225	10.5	16.7	84.4	
*Narasin*	2	2.85	0.10	0.28	1 ^(b)^	12.3	12.9	106.6	0.9996
					2	12.0	12.6	104.0	
					3	11.5	11.8	94.0	
*Salinomycin*	3	3.60	0.59	0.75	1.5 ^(b)^	10.9	13.0	105.6	0.9932
					3	13.3	14.2	92.7	
					5	12.0	12.3	88.3	
*Monensin*	2	2.50	0.22	0.66	1 ^(b)^	13.2	13.6	99.7	0.9995
					2	12.1	12.4	95.6	
					3	11.9	12.4	98.6	
*Maduramicin*	12	13.5	0.24	0.63	6 ^(b)^	13.8	16.9	87.5	0.9979
					12	20.0	21.2	109.6	
					18	20.6	22.2	114.5	
*Robenidin*	25	28.3	1.32	2.09	2.5 ^(a)^	16.4	18.1	95.1	0.9953
					25	12.8	14.1	94.4	
					38	10.3	11.3	104.6	
*Halofuginone*	6	6.58	0.31	0.48	3 ^(b)^	12.6	18.2	86.7	0.9960
					6	13.2	15.8	91.4	
					9	15.6	19.4	88.2	
*DNC*	300	341	2.16	2.76	30 ^(a)^	10.8	11.7	99.0	0.9993
					300	3.7	10.6	88.9	
					450	13.8	18.3	84.6	
*Diclazuril*	2	2.23	0.17	0.20	0.2 ^(a)^	20.4	22.7	86.9	0.9961
					2	17.2	18.9	92.3	
					3	16.9	20.5	99.1	

^(a)^ a validation level of 0.1 MRL; ^(b)^ a validation level of 0.5 MRL.

**Table 3 foods-12-01225-t003:** Frequency (%), mean levels (μg kg^−1^), and range of concentrations (μg kg^−1^) of coccidiostats in samples obtained from different providers.

Coccidiostats	Home-Raised (*n* = 28)					Supermarket (*n* = 34)					Total (*n* = 62)				
	Positive Samples (>MRL)	Frequency	Mean ± SD	Min	Max	Positive Samples (>MRL)	Frequency	Mean ± SD	Min	Max	Positive Samples (>MRL)	Frequency	Mean ± SD	Min	Max
Diclazuril	5 (3)	17.9%	0.91 ± 2.78	0.09	13.6	ND	ND	ND	ND	ND	5 (3)	8.1%	0.46 ± 1.90	0.09	13.6
Nicarbazin	24 (11)	85.7%	266.3 ± 169.4	1.08	744.8	31 (0)	91.1%	167.6 ± 62.2	1.08	222.4	55 (11)	88.7%	212.1 ± 131.3	1.08	744.8
Total	25 (13)	89.3%	133.6 ± 178.9	0.09	744.8	31 (0)	91.1%	83.8 ± 95.0	1.08	222.4	56 (13)	90.3%	106.3 ± 140.9	0.09	744.8

ND—not detected.

**Table 5 foods-12-01225-t005:** Comparison of ADI and EDI (mg kg^−1^ day^−1^), using the average detected concentrations for the children, teenagers, and adult populations.

Compound	ADI (mg^−1^ Kg^−1^ Day^−1^)	Children		Teenager		Adult	
		EDI (mg^−1^ Kg^−1^ Day^−1^)	EDI/ADI (%)	EDI (mg^−1^ Kg^−1^ Day^−1^)	EDI/ADI (%)	EDI (mg^−1^ Kg^−1^ Day^−1^)	EDI/ADI (%)
Diclazuril	0.03	0.00049	1.63	2.25 × 10^−4^	0.75	1.76 × 10^−4^	0.58
Nicarbazin	0.9	0.23	25.5	0.1	11.1	0.08	8.8

ADI—acceptable daily intakes; EDI—estimated daily intake.

**Table 6 foods-12-01225-t006:** Comparison of ADI and EDI (mg kg^−1^ day^−1^), using the highest detected concentrations (i.e., worst-case scenario), for the children, teenagers, and adult populations.

Compound	ADI (mg kg^−1^ Day^−1^)	Children		Teenager		Adult	
		EDI (mg kg^−1^ Day^−1^)	EDI/ADI (%)	EDI (mg kg^−1^ Day^−1^)	EDI/ADI (%)	EDI (mg kg^−1^ Day ^1^)	EDI/ADI (%)
Diclazuril	0.03	0.015	50	0.0067	22.3	0.005	16.6
Nicarbazin	0.9	0.8	88.9	0.36	40	0.28	31.1

ADI—acceptable daily intakes; EDI—estimated daily intake.

## Data Availability

The data presented in this study are available on request from the corresponding author.

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
