# Peer review of "Risk Assessment of Nine Coccidiostats in Commercial and Home-Raised Eggs"

_foods, 2023, doi:10.3390/foods12061225_

Round 1
Reviewer 1 Report
Title: Risk assessment of nine coccidiostats in commercial and home 2 raised eggs
ID No.: foods-2242128
Coccidiosis is one of the major threats to poultry industry worldwide. Coccidiostats are widely used in poultry to combat the disease and thus increase the profit. Use of coccidiostats may lead to resistance development in causal agents as well as transfer of residue through meat and eggs which may adversely affect the health of consumers. Therefore, study of residues of coccidiostats is an important area of research. The manuscript described the potentials of eggs as source of residue source. Chicken eggs were collected from home raised chicken and from supermarket and analyzed for nine coccidionstats by different relevant methods. Method validation was also performed.
General Comments
In general the manuscript is well presented. However, corrections are suggested.
Specific Comments
Specific corrections are directly mentioned in the PDF file.
===
Comments are made on the attached PDF version of manuscript.

Author Response
Thanks for your consideration. Herewith, I send you my responses to the reviewers’ comments to the manuscript " Risk assessment of nine coccidiostats in commercial and home raised eggs" (Foods-2242128). I hope that I had answered to all of the questions and revised the manuscript accordingly but, if not, I would be very grateful for your further observations.
I am looking forward to your reply.
Reviewer 1:
The authors agreed with most of the reviewer’s suggestions and the manuscript was corrected accordingly. Regarding the question made and the removal of Table 4, the authors present their opinion on these subjects.
How the detection frequency was 90.3%? Not clear to me.
The table was corrected since there was a mistake in the total positive samples in home raised samples. There were 25 positive samples in home raised samples and 31 in supermarkets, with a total of 56 positive samples.
Table 4 can be deleted as it is already described in line numbers 270-317
Although part of the data presented in the table is referred to in the text, the table presents additional information. The authors also believe that the systematization in the table helps in the data interpretation. Moreover, it can be useful for other authors seeking this information.
Reviewer 2 Report
This manuscript investigates the risk assessment through the determination of presence of coccidiostats in home raised and supermarket egg samples. This work is commendable but there are some missing information (elaboration) on the data/ findings that the readers may not be able to follow the flow and interpret the data correctly. Hence, a few comments are suggested for the authors to revise the content.
1) The authors are advised to proofread the paper to improve its readability.
2) Abstract should summarize the findings in a way that readers should be able to understand. Please rephrase line 27-30, so readers can understand the comparison of the percentages is coming from which part/sample.
3) Line 20 & 51, please italicize Eimeria.
4) Methodology
Please provide more info on the sampling criteria.
Line 91, stirred and frozen? Can the authors rephrase this sentence to avoid confusion?
Line 121, is the sentence complete? If yes, please put a full stop.
Line 139-140, incomplete info.
For risk assessment, any reason the elderly group is excluded?
In general, some methods require amendment as some descriptions are slightly lengthy and confusing.
Would be good to provide equations for the calculations of parameters in Table 2, and elaborate the validation levels selected/presented.
5) Results & discussions:
Out of 9 compounds, any reason why only diclazuril and nicarbazin are reported? Please elaborate.
line 228, incomplete info, missing (a).
Table 4, for comparison with recent studies findings worldwide, would be good to include data from present study. What can be concluded from this comparison? Elaborate.
Author Response
Thanks for your consideration. Herewith, I send you my responses to the reviewers’ comments to the manuscript " Risk assessment of nine coccidiostats in commercial and home raised eggs" (Foods-2242128). I hope that I had answered to all of the questions and revised the manuscript accordingly but, if not, I would be very grateful for your further observations.
I am looking forward to your reply.
This manuscript investigates the risk assessment through the determination of presence of coccidiostats in home raised and supermarket egg samples. This work is commendable but there are some missing information (elaboration) on the data/ findings that the readers may not be able to follow the flow and interpret the data correctly. Hence, a few comments are suggested for the authors to revise the content.
1) The authors are advised to proofread the paper to improve its readability.
The authors revised the manuscript to improve its readability.
2) Abstract should summarize the findings in a way that readers should be able to understand. Please rephrase line 27-30, so readers can understand the comparison of the percentages is coming from which part/sample.
The authors agreed with the reviewer’s suggestion and the manuscript was corrected accordingly.
3) Line 20 & 51, please italicize Eimeria.
The authors agreed with the reviewer’s suggestion and the manuscript was corrected accordingly.
4) Methodology
Please provide more info on the sampling criteria.
The authors agreed with the reviewer’s suggestion and the manuscript was corrected accordingly.
Line 91, stirred and frozen? Can the authors rephrase this sentence to avoid confusion?
The authors agreed with the reviewer’s suggestion and the manuscript was corrected accordingly
Line 121, is the sentence complete? If yes, please put a full stop.
The authors agreed with the reviewer’s suggestion and the manuscript was corrected accordingly
Line 139-140, incomplete info.
The authors agreed with the reviewer’s suggestion and the manuscript was corrected accordingly
For risk assessment, any reason the elderly group is excluded?
The authors used children (3–9 years old), adolescents (10–17 years old), and adults (18–64 years old). The elderly were not evaluated since the data on consumption and body weight is similar to adults.
In general, some methods require amendment as some descriptions are slightly lengthy and confusing.
The authors agreed with the reviewer’s suggestion and the manuscript was corrected accordingly. However, the authors believe that validation should be an important part of the manuscript, since it supports the results.
Would be good to provide equations for the calculations of parameters in Table 2, and elaborate the validation levels selected/presented.
The equations used in Table 2 are present in the manuscript. Equations 2, 3 and 4 (lines 192-195). The authors did not fully understand the question about the validation levels, however, the levels were selected based on the criteria already mentioned in lines 204-2011.
5) Results & discussions:
Out of 9 compounds, any reason why only diclazuril and nicarbazin are reported? Please elaborate.
There is no justification for that. Since there is a rotation in coccidiostats usage in the feed, this might change over time. Nonetheless, this commentary was added to the discussion.
line 228, incomplete info, missing (a).
The authors agreed with the reviewer’s suggestion and the manuscript was corrected accordingly
Table 4, for comparison with recent studies findings worldwide, would be good to include data from present study. What can be concluded from this comparison? Elaborate.
The authors agreed with the reviewer’s suggestion and the manuscript was corrected accordingly. The discussion was improved to highlight the importance of data comparison.
Reviewer 3 Report
Dear editor authors
Regarding the revision of the manuscript No. foods-2242128, titled “Risk assessment of nine coccidiostats in commercial and home raised eggs”.
Some comments should be replied.
Comments:
1- It is well known that coccidiosis is a self-limiting disease of young animals, since, after a first exposure to coccidia, immunity develops quickly, protecting the bird against future infections. So, I wondered why the table egg producers use the coccidiostats in feed during laying period??? The use of coccidiostats in rearing period is enough for protecting pullets and for developing immunity that protects mature birds form further coccidial infections. Please add your reply to discussion.
2- I think coccidial vaccines are registered in Portugal, so it is easy for most of table egg producers rearing layers on floor to use the vaccine in the first days to avoid the use of coccidiostats. Please add your reply to discussion.
3- Please mention more details the sample collection, how many egg batches per each domestic producer and supermarket samples were examined.

Author Response
Thanks for your consideration. Herewith, I send you my responses to the reviewers’ comments to the manuscript " Risk assessment of nine coccidiostats in commercial and home raised eggs" (Foods-2242128). I hope that I had answered to all of the questions and revised the manuscript accordingly but, if not, I would be very grateful for your further observations.
I am looking forward to your reply.
Reviewer 3
Dear editor authors
Regarding the revision of the manuscript No. foods-2242128, titled “Risk assessment of nine coccidiostats in commercial and home raised eggs”.
Some comments should be replied.
Comments:
- It is well known that coccidiosis is a self-limiting disease of young animals, since, after a first exposure to coccidia, immunity develops quickly, protecting the bird against future infections. So, I wondered why the table egg producers use the coccidiostats in feed during laying period??? The use of coccidiostats in rearing period is enough for protecting pullets and for developing immunity that protects mature birds form further coccidial infections. Please add your reply to discussion.
The authors acknowledge the reviewer’s comment and the manuscript was corrected accordingly adding the following information in the introduction.
There are several strains that can cause coccidiosis, which means that the immunity for one strain is not enough to prevent the infection. Additionally, since coccidiostats only inhibit, reversibly, some of the life stages, they must be used prophylactically during their lifetime to prevent infection.
Although the disease can be self-limiting it induces weight loss (decreased profit) and other issues regarding animal welfare.
- I think coccidial vaccines are registered in Portugal, so it is easy for most of table egg producers rearing layers on floor to use the vaccine in the first days to avoid the use of coccidiostats. Please add your reply to discussion.
The authors acknowledge the reviewer’s comment and the manuscript was corrected accordingly.
Although vaccines exist, they are expensive and their use is not widespread. Moreover, the best results occur with the concurrent use of vaccines and coccidiostats.
- Please mention more details the sample collection, how many egg batches per each domestic producer and supermarket samples were examined.
The authors agreed with the reviewer’s suggestion and the manuscript was corrected accordingly.